# Supporting a Healthier Takeaway Meal Choice: Creating a Universal Health Rating for Online Takeaway Fast-Food Outlets

**DOI:** 10.3390/ijerph17249260

**Published:** 2020-12-11

**Authors:** Louis Goffe, Nadege S. Uwamahoro, Christopher J. Dixon, Alasdair P. Blain, Jona Danielsen, David Kirk, Ashley J. Adamson

**Affiliations:** 1Open Lab., Urban Sciences Building, Newcastle Helix, Newcastle University, Newcastle-Upon-Tyne NE4 5TG, UK; david.kirk@newcastle.ac.uk (D.K.); ashley.adamson@newcastle.ac.uk (A.J.A.); 2Population Health Sciences Institute, Newcastle University, Newcastle-Upon-Tyne NE1 7RU, UK; N.S.Uwamahoro2@newcastle.ac.uk (N.S.U.); J.Danielsen@newcastle.ac.uk (J.D.); 3School of Engineering, Newcastle University, Newcastle-Upon-Tyne NE1 7RU, UK; c.j.dixon1@newcastle.ac.uk; 4Faculty of Medical Sciences, Translational and Clinical Research Institute, Newcastle University, Newcastle-Upon-Tyne NE2 4HH, UK; alasdair.blain@newcastle.ac.uk

**Keywords:** food literacy, nutrition (or nutritional) literacy, takeaway, fast-food, food ordering, digital platform

## Abstract

Digital food ordering platforms are used by millions across the world and provide easy access to takeaway fast-food that is broadly, though not exclusively, characterised as energy dense and nutrient poor. Outlets are routinely rated for hygiene, but not for their healthiness. Nutritional information is mandatory in pre-packaged foods, with many companies voluntarily using traffic light labels to support making healthier choices. We wanted to identify a feasible universal method to objectively score takeaway fast-food outlets listed on Just Eat that could provide users with an accessible rating that can infer an outlet’s *‘healthiness’*. Using a sample of takeaway outlets listed on Just Eat, we obtained four complete assessments by nutrition researchers of each outlet’s healthiness to create a cumulative score that ranged from 4 to 12. We then identified and manually extracted nutritional attributes from each outlet’s digital menu, e.g., number of vegetables that have the potential to be numerated. Using generalized linear modelling we identified which attributes were linear predictors of an outlet’s healthiness assessment from nutritional researchers. The availability of water, salad, and the diversity of vegetables were positively associated with academic researchers’ assessment of an outlet’s healthiness, whereas the availability of chips, desserts, and multiple meal sizes were negatively associated. This study shows promise for the feasibility of an objective measure of healthiness that could be applied to all outlet listings on Just Eat and other digital food outlet aggregation platforms. However, further research is required to assess the metric’s validity, its desirability and value to users, and ultimately its potential influence on food choice behaviour.

## 1. Introduction

The home delivery and takeaway market contributes significantly to the UK economy. In 2017 it was valued at £7.9 billion [1], with more than a fifth of UK residents ordering a takeaway meal at least weekly [2]. Digital food ordering platforms (DFOPs) such as Just Eat, Uber Eats, Deliveroo, Grubhub, and Doordash aggregate access to a market of predominately independent traders. They are increasingly becoming a habitual component of how we choose and purchase takeaway fast-food. Each platform has millions of active users across multiple territories. For example, Just Eat reported that in 2018, in the UK they processed 122.8 million orders from 12.2 million active users [3], representing an increase of 170% (45.5 million) of orders and 122% (5.5 million) active users from their earliest reported figures for the year ending 2014 [4]. Despite their prominent role in food purchasing, little is known about how they shape and inform our dietary choices.

While not exclusively, takeaway fast-food has broadly been characterised as unhealthy. Analysis of 489 meals from 274 independent traders in North-West England found that the majority of meals were high in regards to portion size, energy, macronutrients, and salt [5]. Exposure to takeaway fast-food outlets during daily life of working aged adults has been associated with greater body mass index (BMI), with those most exposed having an estimated 1.21 greater BMI and likelihood of being obese than those least exposed [6]. Frequent consumption of such food is associated with an increased mean daily energy intake [7], with some hypothesising that it is the high energy density that overrides our appetite control systems triggering over-consumption [8]. The positive association between takeaway outlet density and area deprivation [9] has led some academics to conclude that consumption of takeaway fast-food may contribute to health inequalities in overweight or obesity as well as chronic diseases [10].

A successful DFOP must be user-friendly, but its design does not necessarily consider the user’s well-being. While they make the process of selecting and purchasing the desired cuisine-type frictionless, viewed from the perspective of human-centred design, they do not account for the health of the individual and allow them to flourish [11]. Just Eat charges a commission of 14% (excluding VAT) to each takeaway trader on every order processed [12]; therefore, they are motivated to maximise transactions between traders and consumers. As a result, outwardly they aim for them, the platform owner, to be perceived as neutral with respect to an evaluation of a given takeaway and leave the assessment to customer ratings and reviews. This was evidenced in their initial resistance to actively display an outlet’s Food Hygiene Rating in the UK Government-mandated food safety inspections [13]. Nutrition labelling is mandatory on the majority of prepacked prepared food around the world [14,15], with front-of-packaging nutrition labels shown to support consumers in interpreting and selecting healthier products [16]. Yet despite the weight of evidence suggesting that nutritional labelling in the out-of-home setting could play a role in anti-obesity strategies [17], their application has largely been voluntary [18].

Despite the reluctance on the part of DFOPs to display features that could support increased user well-being [19], as aggregators with millions of users they centralise access to thousands of takeaways presenting a potential opportunity for population-level health improvement. Typically, interventions in this sector focus on the individual outlet [20] and are resource intensive [21]. Upstream interventions have greater potential to improve health [22], particularly if the level of agency required to participate is low for both the takeaway trader and the DFOP user [23]. Vidgen & Gallegos define *food literacy* as the scaffolding that empowers us to protect our diet quality [24]. They detail the components that are required to help with the practicalities of meeting nutritional recommendations. To this end, DFOPs are powerfully placed to support two of these component domains: selection as well as planning and management. DFOPs are the structural intermediary at both outlet and meal selection. They provide centralised access (component 2.1) to multiple providers and through increased transparency could provide details of meal ingredients (component 2.2) and guidance to help platform users judge food quality (component 2.3) to enable decisions that balance their food desires regarding taste and nutrition with their financial resources (component 1.3). Conversely, in entering a physical shop, much of the food choice behaviour has already been made, as a customer is limited to the food items within that establishment. Currently identifying a healthy option on a DFOP is an active decision. It requires an outlet to define its cuisine type as *’healthy’* on the DFOP and for the user to subsequently select this as a desired criteria. Health does not have a standardised measurement, but there is an implicit healthiness value in all food that we consume. Therefore, as it is considered essential that all food businesses are inspected and reported for safety, there may also be value in corresponding reporting for health. While it is possible to distinguish between two or more pizza takeaways by their standard of food hygiene, from the nationally administered Food Hygiene Rating scheme [25], there is no equivalent metric with regard to health.

The challenge is to create a universal rating that is both meaningful and equitable. Traditionally, the most commonly assessed intervention is calorie labelling [26], though their reported impact on reducing energy consumed is limited [27]. This may well be related to our poor nutritional literacy [28]. The provision of this information requires the active engagement of the individual trader. There is good-quality nutritional reporting of many of the commonly listed cuisine types [5], but not exhaustive. Though, if one were to define all listed pizza outlets by the one summary metric for pizzas this would ignore the substantial nutritional variability that exists between outlets [5] and defeat the objective of creating a metric that can distinguish between two or more outlets of the same cuisine type in regards to health. Interventions that ‘signpost’ customers towards, or away, from certain options have an advantage over calorie labelling, as they remove the barrier of nutritional comprehension [29]. Traffic light food labels can be considered a form of signposting and have shown promise in supporting healthier choices [30].

In the absence of nutritional information provided by a given outlet on their food offerings, text analysis of the menu content is the most appropriate method to create a measure of the outlet’s overall healthiness. This requires an assessment of what menu text items have an association with health, the applicability of each item to all cuisine types, and the ability of each item to be routinely collected, e.g., the availability of small portion meals [31], salads, and the number of sugar-sweetened beverages [32].

The availability of heuristics that relate to health are desirable for many when we make food choices [33], yet such features which could support increased well-being are neglected by DFOPs. Our aim was to develop a feasible, universal method to objectively score all takeaway fast-food outlets listed on Just Eat, the UK’s most populous DFOP, that could provide users with an accessible rating that can infer an outlet’s *’healthiness’*.

## 2. Materials and Methods

We identified what variables could be routinely collected from takeaway fast-food outlets listed on the DFOP Just Eat and assessed which were associated with academic researchers’ subjective assessment of the overall healthiness of the food offerings of each listed outlet.

### 2.1. Data Source

We selected the DFOP Just Eat, as it had the greatest number of takeaway fast-food outlets listed on its platform within our setting of the UK. Due to the requirement of an academic assessment of each listing along with the prohibition of data-scraping technologies in accordance with Just Eat’s website terms and conditions [34], we sampled those outlets listed on the platform on November 2018 that served the central Newcastle-upon-Tyne postcode of ’NE1’, a major city in the North-East of England.

### 2.2. Variables

#### 2.2.1. Expert Health Score

We used convenience sampling, due to project time and resource constraints, and contacted academic researchers working in the field of public health and nutrition, via a Newcastle University mailing list of approximately 20 people, to provide their individual assessment of the healthiness of 160 takeaway menu listings on Just Eat that served the NE1 postcode. They were asked to score the menu listings with respect to health as follows: 1, poor; 2, OK; 3, good. We intentionally did not provide researchers with specific criteria as to how to score, but we did provide examples of menu cues that may help support their individual assessment, e.g., availability of meal deals, provision of nutritional information, and portion sizes. As exploratory research, a prescriptive measure based on set criteria would not have captured the range of views associated with perceptions of health. Furthermore, theoretically it was important to identify those menu metrics that were associated with the expert scores and not define the measure prior to analysis. The scores from all researchers for each outlet were combined to create a cumulative score.

#### 2.2.2. Online Menu Metrics

We reviewed Just Eat takeaway listings for potential metrics that could be extracted from each menu listing. They had to fit the following criteria: the ability to be routinely collected through automated processes, had an established association with health, and could be applicable to all listed takeaways. We identified 15, which are listed and detailed in Table 1. Author JD manually extracted the data for each menu listing and metric. Author LG subsequently carried out quality assurance on the data to identify any mistakes.

### 2.3. Data Analysis

We used generalized linear modelling (GLM) to asses the relationship between the available menu metrics and the cumulative expert health score. All 15 menu metrics listed in Table 1 were included as independent variables in the GLM. Due to the exploratory means of our study, limited sample size, and crucial lack of rigorous nutritional information regarding each food item, we were not looking for definitive variables associated with takeaway fast-food health, but rather those that could appropriately contribute to a human-centred metric. As such, significance for independent variables was set to *p* < 0.1 in stepwise deletion. The resultant regression equation from our GLM was then used to derive a fitted score for each outlet and then converted into a Health Rating using the Food Standards Agency (FSA) Food Hygiene Rating Scheme as a template [25].

We carried out data analysis in R [35].

## 3. Results

The complete data can be found in Appendix A. Takeaway outlets have been anonymised, but we have retained each outlet’s listed cuisine tags. The table also contains for each takeaway outlet the expert health score, the score for each of the 15 menu metrics, the GLM fitted score, and their resultant Health Rating. It took approximately two months to collate the expert health scores. During this time, due to the high rate of attrition in the takeaway fast-food sector [36] from the original list of 162 menu listings to review, 149 (92%) were included in the data analysis. Each outlet was scored four times independently. These scores were added together to create a cumulative measure that ranged from a minimum of 4 to a maximum of 12. The significant coefficients from the GLM can be seen in Table 2.

The availability of water as a drink option, salads, and the diversity of vegetables were positively associated with nutrition researchers’ assessment of an outlet’s healthiness, whereas the availability of chips and multiple meal sizes were negatively associated. The number of dessert options was also found to be negatively associated with our expert health score, though this was at *p* < 0.10; therefore, there is an increased likelihood that we obtained this result by chance. However, it was retained in the GLM for the reasons detailed in Materials and Methods.

### 3.1. Model Residuals

In order for our GLM with a Gaussian error structure to be a suitable model for the data, the residuals (error) should be normally distributed with a zero mean. The resultant distribution of the residuals from our GLM was normal.

### 3.2. Health Rating

Using the FSA’s Food Hygiene Rating scheme [25] as a template to rate outlets from 0 to 5, we assigned a score based on their fitted score derived from the GLM equation. As outlets could achieve a minimum of 4 to a maximum of 12 from the cumulative expert health score, we decided on the equally distributed cut-offs across this range, see Table 3.

Figure 1 shows the distribution of takeaway outlets across the assigned Health Rating scores. The most frequent rating was 1, with no takeaway outlets achieving the highest obtainable rating of 5.

In their Just Eat listing, takeaway outlets are tagged with a maximum of three cuisine types (assumed to be defined by the individual trader). Across the 149 outlets there were a total of 47 different cuisine tags. Table 4 shows the distribution of our Health Rating scores by cuisine tags for those tags with at least three takeaway outlets.

The majority of the most popular cuisine types—pizza, halal, kebab, Italian, American/burgers, and Chinese/Oriental—followed the broad trend of frequency of Health Rating peaking at either 0 or 1 and declining, but with some options in the higher score of 3 or 4. Notable exceptions to this pattern were Indian, curry, and Bangladeshi, which all peaked at a Health Rating of 2. Grill had a U-shaped distribution, with examples at either end of the spectrum and none that scored a Health Rating of 2. Lebanese had the most positively skewed distribution, where no outlet scored less than a Health Rating of 3. Fish & Chips were the only savoury cuisine tag not to score over a Health Rating of 1, and while the cuisine tags Desserts and Cakes were broadly low scoring, there was one example of an outlet with a Cake cuisine tag that achieved a Heath Rating of 2.

## 4. Discussion

### 4.1. Summary of Principal Findings

Our work demonstrates that it is feasible to create an outlet-level health metric that suitably segregates takeaway outlets of the same cuisine type. The availability of salads, the diversity of vegetables, and the number of water options contributed to a higher Health Raking score. Such variables are unsurprising and are established components of UK Government dietary recommendations [37]. Deep-fried potatoes, e.g., chips, fries, and wedges, are ubiquitous in the takeaway fast-food sector and were present in all but seven of the outlet menus in our study, see Appendix A. Their increased presence on a menu contributed a lower Health Rating, and their regular consumption has been associated with poor health outcomes [38]. The availability of desserts contributed to a negative Health Rating. Their intrinsic high sugar content is a component of our diet that has been linked to numerous adverse health outcomes including increased body weight [32]. Conversely, the presence of sugar-sweetened beverages was not found to contribute to the Health Rating equation. This was surprising given that such products were the focus of a recent UK Government intervention [39]. Notably, the availability of meal items in multiple sizes contributed to a lower Heath Rating. This may be as a result of the prevalence of large portion meals, particularly for pizzas, as conversely the availability of smaller portions was not found to significantly contribute to the Health Rating equation.

### 4.2. Strengths and Limitations of Study

We do not propose that our resultant regression equation is the definitive algorithm to create a takeaway outlet-level Health Rating score. Rather, it is a provocation that demonstrates it is feasible to create an objective measure that infers a takeaway outlet’s relative healthiness. It is based on variables that have the potential to be routinely collected digitally and applied uniformly across all takeaway cuisine types and to other DFOPs. It shows that there is potential to create a measure that would allow for an individual to filter the potentially healthier takeaway outlets of their preferred cuisine choice.

The fact that the resultant metric is associated with distinct nutritional characteristics, as opposed to identification of broad trends through a dimension-reducing analysis such as principal component analysis, provides a feedback mechanism to those takeaway outlets interested in improving their Health Rating. It encourages greater transparency in reporting of the description of meals by detailing and increasing vegetable content, providing more salad options, and the greater availability of water.

The limited sample size has resulted in an equation that appears to systematically bias against certain cuisine types. No Fish & Chip shops attained a Health Rating above 1. While a larger sample size would be required to determine if the presence of desserts should be included on statistical grounds, currently with this as an explicit component of the Health Rating equation, all dessert-focused outlets are inherently disadvantaged with regards to achieving a higher Health Rating score. This Health Rating only reflects the experience of those ordering takeaway fast-food from central Newcastle-upon-Tyne. An expanded geographical analysis is required to account for the regional variation that exists in takeaway cuisine types as well as the turbulence in outlet ownership [36]. However, to achieve an increased sample size would require the cooperation of the DFOP, as the alternative method of web scrapping to obtain the data violates their website’s terms and conditions.

We did not validate the expert health scores. However, this would be challenging as there is no universal measurement of health. We would likely have to apply one or more *a priori* nutritional metrics such as calorie content of menu items, which would be costly and time consuming to do at an appropriate scale. An alternative method would be to crowd-source user views on an outlet’s health as part of the platform’s automated process in which they request users’ reviews. The resulting scores could contribute to an evolving Heath Rating that employs machine learning techniques.

### 4.3. Comparison with Other Studies and Interpretation of Findings

If such a Health Rating were to be applied, a more nuanced metric is likely to be required. For example, the FSA’s Food Hygiene Rating scheme [25] is a calculation of an assessment of an outlet’s compliance with food hygiene and safety procedures, compliance with structural requirements, and confidence in management/control procedures [40]. If an outlet scores poorly in one category, this limits the overall Food Hygiene rating that they can achieve. For example, if food is hygienically handled and the outlet is in a good condition, but documentation has not been well managed, an outlet cannot achieve an overall high rating [40]. There were three metrics that we collected: smaller portions, healthy/ier options, and nutritional information, which are all explicitly linked to health and the ability to guide and increase an informed choice. However, none of these metrics was significant in our model. In the case of healthy/ier options and nutritional information, neither metric was useful at segregating outlets as there was only a few incidences of these items being reported on menu listings. We only found nine outlets that listed healthy/ier options and only one outlet that provided nutritional information. Despite the evidence that the large portion sizes are an established component of UK takeaway fast-food [5], the widespread availability of smaller portions was not associated with our expert health scores. While it would not be acceptable that takeaway outlet owners make wholesale changes to their cuisine types to ones that are healthier, these three factors of smaller portions, healthy/ier options, and nutritional information could and should be encouraged. Therefore, a more complex algorithm that provides a score boost, for example +1 for the provision of calorie labelling for meal items, could be included. For example, the only outlet that provided nutritional information for all its food offerings achieved a relatively low Health Rating of 2, markedly lower than the cumulative expert health score of 9 out of a possible 12.

Early in the implementation of the FSA’s Hygiene Rating scheme, independent evaluation reported that consumers supported the scheme in principle, stating that an objective indication of food hygiene standards was useful [41]. However, consumers at that time stated they lacked the understanding on how to interpret the rating and that engagement with it was related to an individual’s position regarding the importance of hygiene. The report stated that consumer interest in the scheme would increase over time as awareness and understanding improved [41]. In 2019, public awareness and interest in the scheme was born out following a BBC report, detailing that DFOPs had listed hundreds of takeaways with poor FSA Hygiene Ratings [19]. This resulted in platforms modifying their practices to display each outlet’s Hygiene Rating [13]. This suggests that implementation of a Health Rating would take time to gain consumer interest, but would likely be of benefit if its structure was closely related in format to the FSA’s Food Hygiene Rating scheme, as this would reduce the comprehension cost. At this point of theorising of a potential metric, there is no guarantee of user interest, as while the availability of a health heuristic may be desirable broadly when it comes to food choice [33], it is unknown if health is a critical consideration when specifically purchasing takeaway fast-food. Furthermore, it is unclear how a Health Rating might impact on health inequalities. Vegeris’s evaluation of the FSA’s Hygiene Rating scheme stated that consumers who were more deliberate in their food choice were more likely to refer to the Hygiene Rating. Therefore, a Health Rating may only be of interest to those with a particular interest in healthy food. Conversely, it may reduce health inequalities, as it is in effect an intervention delivered exclusive to those purchasing takeaway fast-food.

### 4.4. Implications for Policy and Practice

In 2018 the UK Government consulted on its intention to introduce mandatory calorie labelling in the out-of-home food sector in England [42], which has the potential to lead to public health benefits [43]. It stated that it would legislate for such labelling for businesses with 250 or more employees, which would exclude the vast majority of businesses listed on DFOPs. Feedback stated that there was a concern that small businesses would find calculation and provision of calories particularly challenging [42], despite freely accessible online recipe analysis tools such as MenuCal [44]. The application of a Health Rating score would require only the engagement of the DFOP either voluntarily or by legislative means as opposed the active participation of every takeaway outlet trader. While menu calorie labelling offers a more robust measure of a food item’s relative healthiness, regulation would also have to be considered. In the UK, the responsibility would likely fall on either local authority trading standards or environmental health teams, who have stated that existing delivery of food hygiene inspections are time and resource intensive [45]. In 2019, the total number of takeaway fast-food shops and mobile food stands in the UK was estimated to be in excess of 40,000 [46]; therefore, alongside restaurants the number of meals that would require calorie labelling would number in the hundreds of thousands, if not millions. It would require complex and costly systems to ensure that meals were accurately reported. A Health Rating inline with our proposal would be cheap and universal and could also work either as an abridge to future calorie labelling or in tandem. It could also support the UK Government’s stated objective to “Empowering everyone with the right information to make healthier choices", as part of their strategy to tackle obesity published in 2020 [47].

The resistance shown by DFOPs to display Hygiene Ratings suggests that governments may be required to legislate for the provision of a Health Rating. However, if implemented, it should be accompanied with transparent information to takeaway fast-food business owners as to how the metric is constructed and guidance as to how they could improve their business’ score. This would provide the feedback mechanism that could support an outlet-level improvement in the healthiness of their offerings. It would also require the support of each DFOP to assist outlets in updating their menu listing, as often modifications incur a cost.

### 4.5. Unanswered Questions and Future Research

It is unknown if DFOP users would want and value a Health Rating, as well as if its provision would influence outlet and food choice. A 2016 systematic review on the impact of interventions to promote healthier eating in out-of-home outlets found that both signposting of healthier options and calorie labelling may have positive intervention effects; however, the evidence was limited, and the included studies were of low or moderate quality [29]. This may be a result of low nutritional literacy with respect to consumer comprehension of calories and their respective energy intake [28]. Or it might be that health is not a primary consideration when choosing to eat takeaway fast-food.

Delivering this research would most likely require A/B testing and the partnership of a DFOP. It is unknown how likely they would be to engage in such work, where the implications may not be seen as desirable to the listed takeaway outlets. There is also strong opposition from some academics regarding whether we should accept industry funding of public health research as their influence biases science [48]. Alternative methods may utilise web-augmentation techniques [49], though further work is required to see if this is feasible. Design research would also provide an insight as to how DFOP users interpret the Health Rating scores. As identified in the evaluation of the FSA Hygiene Rating scheme [41], consumers may need additional information to understand the differences between and meanings of each score.

While there is currently no universal health metric, there are other forms of signposting that allow outlets to explicitly define their cuisine type through tags and the provision of nutritional metrics such as calories. Further work could be done to see how such outlets would be rated using our proposed measure.

It would also be essential to identify and understand any unintended consequences and a Health Rating scheme’s role in health inequalities. For example, for many the choice of a takeaway fast-food meal is a treat and health is not a consideration as exemplified by the rise in competitive eating [50] and food challenges [51]. Therefore, a Health Rating might be used to find an explicitly unhealthy outlet.

## 5. Conclusions

The past decade has seen a marked shift towards digitally mediated methods of food ordering, a trend that is likely to continue. While it is unknown if DFOPs are increasing our consumption or just providing an alternative mode of access, they do provide a centralised access point which allows for potentially population-level public health interventions that support informed and healthier food purchases. A Health Rating based on online menu listings is feasible, but as yet we do not propose that its implementation would be a panacea to guiding healthy choices. More research is required to understand how it could influence both the DFOP user and takeaway outlet behaviour. However, it highlights what more DFOPs, like Just Eat, could provide to their users to make their technology more human-centred and support an improvement in food literacy.

## Figures and Tables

**Figure 1 ijerph-17-09260-f001:**
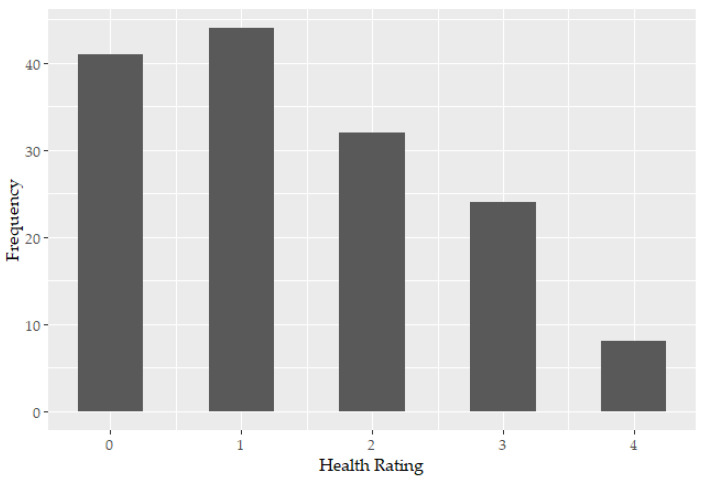
Frequency of takeaway outlets per each Health Rating score within the study sample.

**Table 1 ijerph-17-09260-t001:** Just Eat online menu items associated with health that are identifiable and quantifiable.

Metric	Description
Special offers	The number of special offers, meal deals, sharing meals, set meals
Multi-size item	The number of meals that are available in multiple sizes
Smaller portions	The number of meal items that are specifically labelled as small
Healthy/ier options	The number of items with a health-related label, e.g., lite, clean
Desserts	The number of dessert items, e.g., cake, sweets, ice cream
Salads	The count of all mentions of salad or related items, e.g., coleslaw
Chips	The count of all mentions of chips/fries/wedges
Vegetables	The number of different vegetables mentioned (not fruit)
Dietary requirements	The number of different dietary requirements explicitly catered for
Nutrition information	The number of nutritional metrics, e.g., kcal
Sugar-sweetened beverages	The number of sugar-sweetened beverages
Diet drinks	The number of diet drinks
Milk	The number of milk (non-sweetened) options
Water	The number of water options
Alcohol	The number of alcoholic drinks

**Table 2 ijerph-17-09260-t002:** Significant coefficients from the generalized linear model associated with the nutrition researchers’ cumulative health score, reporting each independent variable’s estimate, standard error, *t*-test, and *p*-value.

Coefficient	Estimate (Std. Error)	t Value	Pr (>|t|)
Intercept	5.57 (0.40)	13.85	<0.01
Multi-size item	−0.03 (0.01)	−4.76	<0.01
Desserts	−0.03 (0.02)	−1.75	0.08
Salads	0.08 (0.02)	4.41	<0.01
Chips	−0.05 (0.01)	−3.52	<0.01
Vegetables	0.17 (0.04)	4.49	<0.01
Water	0.40 (0.16)	2.44	0.02

**Table 3 ijerph-17-09260-t003:** Health Rating and their associated cut-offs from the GLM fitted scores.

Health Rating	Fitted Score Cut-Off
0	<5.33
1	5.33–6.66
2	6.66–7.99
3	7.99–9.33
4	9.33–10.66
5	>10.66

**Table 4 ijerph-17-09260-t004:** Frequency of outlets for each assigned Health Rating by cuisine tag.

	Health Rating					
Cuisine Tag	0	1	2	3	4	5
Pizza	27	22	8	2	2	0
Halal	18	17	10	9	2	0
Indian	3	6	19	3	0	0
Kebab	8	9	1	3	0	0
Italian	6	7	3	2	2	0
Grill	4	2	0	6	3	0
Curry	0	2	7	3	0	0
Fish & Chips	9	3	0	0	0	0
American	5	4	0	2	0	0
Lebanese	0	0	0	9	2	0
Burgers	2	6	0	1	1	0
Chicken	5	4	1	0	0	0
Chinese	2	4	3	1	0	0
Oriental	1	4	1	2	0	0
Bangladeshi	0	1	4	0	0	0
Desserts	3	2	0	0	0	0
English	2	2	1	0	0	0
Breakfast	0	3	1	0	0	0
Cakes	2	0	1	0	0	0

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
