# Peer review of "Supporting a Healthier Takeaway Meal Choice: Creating a Universal Health Rating for Online Takeaway Fast-Food Outlets"

_ijerph, 2020, doi:10.3390/ijerph17249260_

Round 1

Reviewer 1 Report

This paper is very well written and the topic is great! Yet, there seems to be some difficulty with the entirely subjective nature of the process that is then framed as creating an “objective” measure. This creates problems with the manuscript, which need to be addressed.

The authors spend much time explaining that there is a problem with nutritional reporting; that, for example, the nutritional quality of pizza from one outlet might be different from another, and that one needs to be able to distinguish between two outlets with the same cuisine. They then proceed to say that “text analysis of the menu content the most appropriate method”.  The claim is unconvincing. How does text analysis reliability reveal the “health” of one pepperoni pizza over another? I understand what the authors are aiming to do, but to put it frankly, adding water to the menu doesn’t make the outlet as whole any healthier.

The subjective assessment that underpins this needs to be revisited, and for the purposes of scientific rigor, one would assume the metric should be validated prior to publication. As per the methods: “we… assessed which were associated with academic researchers’ subjective assessment of the overall healthiness of the food offerings”. The limitations that the researchers themselves identify are spot on, including “the limited sample size and crucially lack of rigorous nutritional information regarding each food item” and the fact that the metric has not been validated or assessed in terms of its desirability or value to users.

Moreover, the subjective assessments that are reported – when it comes to the ‘findings’ -- are not surprising: it is expected that including water as a drink option and offering salad and a diversity of vegetables would be positively associated with nutrition researchers’ assessment of an outlet’s healthiness while the availability of chips is negatively associated. I agree with the authors that this is a provocation – but the claim that it is creating an “objective measure” with respect to a takeaway’ outlet’s relative healthiness is problematic, given that they authors are explicit throughout about the fact that the measure is purely subjective.

 Certain points require clarification – i.e., lines 69 – It says it is possible to distinguish between two or more pizza takeaways by their hygiene rating but it is not possible with regards to health. Why not? Is there a “hygiene” rating that consumers provide? Is it possible for consumers to also provide a perceived “health” rating? This needs to be explained – in terms of what a DFOP provides//

On Lines 30 – 31 the article claims, “Despite their prominent role in food purchasing, little is known about how [DFOP] shape and inform our dietary choices”. This seems to suggest that the research will provide some insight into this gap of knowledge.

Citing Vidgen & Gallegos’ food literacy as “the scaffolding that empowers us to protect our diet quality” does not logically connect to the claim about DFOP made in the subsequent sentence. If this is about food literacy, then it needs to be engaged with in some meaningful way. More is required in terms of exploring why DFOP is a matter of food literacy (?!) and why the claim is valid.  As it stands, this functions as merely a throwaway quote.

Grammatical issues:

Sentence fragment  - line 80

Faulty parallelism – lines 108-110

Reviewer 2 Report

The manuscript is undoubtedly interesting and may be of interest to potential readers. The work raises an interesting problem, it is well written. However, what disturbs me mainly in the reception of the work is the lack of detailed descriptions of tables and figures. These descriptions should make it possible to understand the contents of the tables and figures without reading the entire work in detail. The symbols and markings used should be explained in the tables and the drawing. Certainly, the authors should consider this aspect of their presentation. Additionally, in my opinion, citation 44 should be removed or repleaced by by note or a well-discussed comment.

Reviewer 3 Report

The authors presented a paper about developing a rating system for foods listed on a digital ordering system. Overall, this was an interesting paper, but the reviewer has a few comments to strengthen it:

Abstract: Clarify if this is globally or within a certain country that people use a digital ordering platform system. “Unhealthy” is not a globally defined term, thus are these foods considered non-nutritious? Or contain too many nutrients that would lead to non-communicable diseases? Clarify how many nutrition researchers were included. What was the minimum/maximum scores? Please include statistics as associations is a broad term.

Introduction:

As restaurants now include calories on their menus or at least indicators to signify it is ‘healthy’, it really has not helped reduce the number of individuals who continue to order high caloric foods, the number of people who continue to eat out nor reduce obesity. Therefore, it would be good to include this information to explain the rationalization as to how this digital ordering rating system would help compared to the standard menus. It would be also good to mention if in the UK, there is use of the traffic light system on the food packages, how this device would use the same system for consistency. Please define healthy.

Line 32: Define unhealthy

Lines 33-34: Explain why consumption of fast food increases BMI. Include the frequency of consuming these foods and the types of foods that are ordered that increase this risk. For example, ordering a plain small hamburger likely has lower odds of one becoming obese compared to ordering large fries, double cheeseburger and a sugar-sweetened beverage.

Methods:

Line 99+: Indicate the number of researchers you reached out to and a rationale for that number. Include the rationalization for not informing a researcher how to score. Since you did not instruct these researchers of how to score these items, did you include why they rated an item a certain way? Mentioned how the items were scored, but for each item and then averaged? Further clarify.

Explain the variables included within the GLM.

Results:

Table 4. This is slightly confusing with including the total column. It should be eliminated because the focus was on the rating from 0-5 with 5 indicating healthy and 0 unhealthy. The total makes it appear that pizza was the healthiest with cakes identified as the least healthy.

Discussion:

Even though few if any digital ordering systems include rating scores based on a product’s ‘healthiness’, it would be good to include how the ratings matched up with ratings of products that had a nutrition facts label. For example, apple slices compared to fruit snacks.

Line 172: clarify that the high sugar is simple sugars or added sugars.

Round 2

Reviewer 3 Report

The authors have substantially improved this manuscript. There were a few areas that needed a bit more clarity:

  • Line 32 in the introduction, it is the opinion of others that fast food establishments are unhealthy, but expand on this meaning fast food places generally offer high sodium/high fat/added sugar meals as opposed to whole grains, fruits and vegetables. Then include references for that statement.
  • Lines 33-34 in the introduction, what is meant by greater BMI and odds of obesity? Some numbers would help clarify the greater part.
  • Lines 55-56: include a reference(s) as the above information in that paragraph did not indicate that DFOP is reluctant to share health information about the meals.
  • Lines 68-72 need references as it reads more opinion based.
